# Evolution of metazoan oxygen-sensing involved a conserved divergence of VHL affinity for HIF1α and HIF2α

Daniel Tarade [1], Jeffrey E. Lee[1] & Michael Ohh [1,2]

Duplication of ancestral hypoxia-inducible factor (HIF)α coincided with the evolution of vertebrate species. Paralogs HIF1α and HIF2α are the most well-known factors for modulating the cellular transcriptional profile following hypoxia. However, how the processes of natural selection acted upon the coding region of these two genes to optimize the cellular response to hypoxia during evolution remains unclear. A key negative regulator of HIFα is von Hippel-Lindau (VHL) tumour suppressor protein. Here we show that evolutionarily-relevant substitutions can modulate a secondary contact between HIF1α Met561 and VHL Phe91. Notably, HIF1α binds more tightly than HIF2α to VHL due to a conserved Met to Thr substitution observed in the vertebrate lineage. Similarly, substitution of VHL Phe91 with Tyr, as seen in invertebrate species, decreases VHL affinity for both HIF1α and HIF2α. We propose that vertebrate evolution involved a more complex hypoxia response with fine-tuned divergence of VHL affinity for HIF1α and HIF2α.

[1] Department of Laboratory Medicine & Pathobiology, University of Toronto, 1 King's College Circle, Toronto, ON M5S 1A8, Canada. [2] Department of Biochemistry, University of Toronto, 661 University Avenue, Toronto, ON M5G 1M1, Canada. Correspondence and requests for materials should be addressed to M.O. (email: michael.ohh@utoronto.ca)

Metazoan oxygen-sensing requires hypoxia-inducible factor (HIF)α proteins[1]. These oxygen-labile transcription factors are present at low to non-detectable levels under normoxic conditions in metazoan cells but are stabilized at the protein level under hypoxic conditions. In a series of experiments, it was shown that HIFα protein is rapidly hydroxylated on conserved proline residues found within its oxygen-dependent degradation (ODD) domain by prolyl hydroxylase domain containing enzymes (PHDs)[2]. The hydroxylation of the proline residues allows recognition of HIFα by von Hippel-Lindau (VHL) protein, which serves as the substrate-conferring component of an E3 ubiquitin ligase complex[3–5]. The hydroxylation of HIFα via PHDs requires molecular oxygen as a co-substrate. Thus, hydroxylation of HIFα results in oxygen-dependent, ubiquitin-mediated, proteasomal degradation. Under hypoxic conditions, HIFα escapes both hydroxylation via PHDs and recognition via VHL allowing its dimerization with the constitutively expressed HIFβ protein. This dimerization results in the formation of a functional transcription factor that upregulates expression of angiogenic factors, glycolytic enzymes, and signallers of erythropoiesis, among other genes.

All eumetazoans, except ctenophera, express, at minimum, a single HIFα, PHD, and VHL gene[6]. The simplest metazoan to express the full complement of PHD, VHL, and HIFα genes is T. adhaerens[7]. PHD cloned from T. adhaerens can function to regulate HIFα when expressed in human cells, suggesting remarkable conservation of this pathway during metazoan evolution[7]. Crystallization of T. adhaerens PHD bound to its corresponding HIFα peptide reveals significant structural conservation with human PHD2 bound to human HIFα peptides[8]. However, despite conservation of the core proteins critical for oxygen-sensing throughout the majority of animal evolution, duplication of the genes encoding these proteins is observed. All examined invertebrate species possess only one HIFα gene and at most two PHD genes[7]. Interestingly, HIFα is observed to have undergone multiple duplication events coinciding with the evolution of vertebrate species[7].

In humans, there are three HIFα proteins, HIF1α, HIF2α, and HIF3α, among which the former two are best characterized. There are considerable similarities between HIF1α and HIF2α proteins, which have 48% primary amino acid sequence identity[9]. The primary hydroxylation sites are P564 and P531 in HIF1α and HIF2α (C-terminal oxygen-dependent degradation domain or CODD sites), respectively, while the corresponding secondary hydroxylation sites are P402 and P405 (N-terminal ODD or NODD sites). Structural data obtained from co-crystallization of HIF1α and HIF2α CODD peptides with the VHL regulatory complex reveal a similar binding motif for the two protein homologs[10–12]. There is also high structural conservation between HIF1α and HIF2α basic helix loop helix (bHLH) and PER-ARNT-SIM (PAS) domains, which are responsible for DNA binding and dimerization with HIFβ, respectively[13].

Although HIF1α and HIF2α are regulated by PHDs and VHL in a similar manner, several differences exist between the two proteins. Firstly, HIF1α and HIF2α are not redundant during mouse embryogenesis. HIF1α$^{-/-}$ mice are embryonic lethal (E11) as a result of defects in neural tube and cardiovascular development[14,15]. HIF2α$^{-/-}$ mice also die during embryogenesis due to defects in catecholamine synthesis or vascular disorganization[16,17]. However, the penetrance and timing of embryonic lethality appears to depend on the genetic background[16,17]. Furthermore, despite the fact that HIF1α and HIF2α both recognize the canonical hypoxia-responsive element (HRE; 5′-RCGTG-3′) and regulate a common set of genes, each also regulates a set of unique transcriptional targets[18,19]. For example, regulation of genes encoding glycolytic enzymes is mostly carried out by HIF1α whereas HIF2α is the near-exclusive binder of HREs associated with members of the Oct4 family[18]. Additionally, although HIF1α and HIF2α both appear to contribute to the pathogenesis of clear cell renal cell carcinoma (ccRCC) in the context of VHL loss, they appear to play different roles. A series of experiments with transgenic mice show that HIF1α but not HIF2α activation results in metabolic reprogramming and promotion of the clear cell phenotype that is important for tumour initiation[20–22]. Conversely, HIF2α has been shown to be necessary for the growth of xenograft tumours[23,24]. Lastly, HIF1α responds acutely to hypoxia while HIF2α is largely a chronic responder to hypoxia due in part to its less efficient degradation[25]. Despite well-documented differences between HIF1α and HIF2α, how natural selection acted upon the coding region of these two genes during the evolution of vertebrate species has only been studied using computational methods[26]. We sought to identify substitutions that uniquely impacted one of the HIFα paralogs and to evaluate the potential impact on protein function, which may have facilitated increased complexity of the oxygen-sensing pathway.

Here, we show that HIF1α binds more tightly to VHL than HIF2α due to a conserved substitution in the vertebrate lineage affecting an amino acid proximal to the primary hydroxylation site. HIF1α Met561 but not HIF2α Thr528 stabilizes the interaction with VHL by favourably packing against VHL Phe91. Similarly, VHL F91Y substitution, which mimics a common invertebrate substitution, results in decreased VHL affinity for both HIF1α and HIF2α. Thus, we propose that following the emergence of the vertebrate clade, these conserved substitutions resulted in a functional divergence in VHL-mediated negative regulation of HIF1α and HIF2α, allowing more specialized hypoxic signalling.

## Results

**HIF1α has a higher affinity for VHL than HIF2α.** The relative affinities of VHL for the two major paralogs of HIFα have not been previously reported. As it has been suggested that HIF2α is less efficiently degraded when compared to HIF1α, we hypothesized that HIF1α may interact more strongly with VHL. In a pulldown assay, we observed that hydroxylated HIF1α peptide (556–573) pulls down more in vitro transcribed and translated (IVTT) HA-VHL than HIF2α peptide (523–541; Fig. 1a). Furthermore, acetylated HIF1α peptide competes more effectively for HA-VHL than HIF2α peptide (Fig. 1b). These results suggest that HIF1α binds more tightly to VHL than HIF2α.

**HIF1α Met$_{n-3}$ stabilizes interaction with VHL.** We have previously noted that in the immediate vicinity of the primary hydroxylation sites there are several residues that are not conserved between HIF1α and HIF2α[12], including the amino acid three residues N-terminal of the primary hydroxylation site, HIF1α Met$_{n-3}$ and HIF2α Thr$_{n-3}$ (n denotes the primary hydroxylation site; Fig. 2a). We hypothesized that this difference in affinity for VHL might be mediated by the divergent biochemical properties of HIF1α Met$_{n-3}$ and HIF2α Thr$_{n-3}$. To test this hypothesis, we designed a HIF1α hybrid screen whereby residues that are not conserved between HIF1α and HIF2α are substituted to the corresponding HIF2α residue (Fig. 2a). Indeed, HIF1α M561T pulled down similar levels of HA-VHL when compared to HIF2α wild-type (WT), while other substitutions designed to mimic HIF2α, including insertion of Gly$_{n+6}$, had no effect on affinity for VHL (Fig. 2b). Substitution of HIF1α Met561 with alanine was also found to reduce affinity toward VHL to levels similar to HIF2α WT (Supplementary Fig. 1a). Importantly, the reciprocal HIF2α T528M substitution resulted in increased

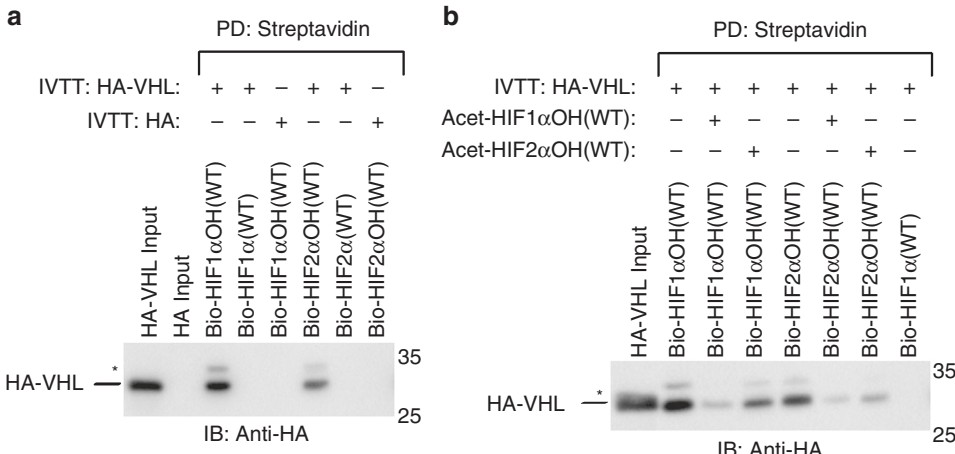

**Fig. 1** VHL has a higher affinity for HIF1α than HIF2α. **a**, **b** Biotinylated HIFαOH peptides were immobilized on streptavidin- agarose beads and incubated with in vitro transcribed and translated (IVTT) HA-VHL. Where indicated, acetylated HIFαOH peptides were included during the pull down. Streptavidin beads were pulled down (PD) and levels of HA- VHL were visualized via immunoblotting (IB). Molecular weight markers (kDa) are labeled for western blots

affinity for VHL while deletion of G537 resulted only in a minor decrease in affinity for VHL (Supplementary Fig. 1b). To further evaluate the effect of Met/Thr$_{n-3}$ on HIFα/VHL complex stability, we performed biolayer interferometry (BLI), where the rate at which purified VHL-elongin B-elongin C (VBC) complex associates and dissociates to and from immobilized HIFα peptide can be measured (Supplementary Fig. 2a). We found that Met$_{n-3}$, as found in HIF1α WT, is associated with a dissociation rate that is 1.7 to 1.8 times slower than peptides containing Thr$_{n-3}$ (HIF2α WT; Fig. 2c, d and see BLI sensorgrams in Supplementary Fig. 2a). The kinetic values were validated using surface plasmon resonance (SPR; Supplementary Fig. 2b). We found that HIF1α WT had a dissociation rate ($0.024 \pm 0.0007\,s^{-1}$) that was slower than both HIF2α WT ($0.067 \pm 0.004\,s^{-1}$) and HIF1α M561T ($0.076 \pm 0.01\,s^{-1}$; Supplementary Fig. 2c). Next, immunoprecipitation of IVTT 3xFLAG-HIF1α(387–581) revealed that the M561T substitution resulted in decreased affinity of HIF1α ODD protein for VHL while the T528M substitution increased the affinity of 3xFLAG-HIF2α(390–554) ODD protein for VHL (Fig. 2e, Supplementary Fig. 1c). Thus, we find that non-conserved residues in the proximity of the primary hydroxylation site of HIF1α and HIF2α dictate different binding affinity to VHL.

**HIF1α Met$_{n-3}$ is conserved in vertebrate species**. Having shown that HIFα Met$_{n-3}$ increases affinity for VHL, we next asked whether this residue is conserved in metazoan species. First, a maximum-likelihood phylogenetic tree was constructed using the cDNA sequences corresponding to the ODD domain of HIF1α (Fig. 3a). The maximum-likelihood method attempts to minimize the number of evolutionary events required to explain the sequence data while also allowing for variable rates of evolution across sites and branches. Generally, known relationships between major clades of metazoan species are recovered in our experimental phylogenetic tree. Subsequently, we estimated synonymous (dS) and non-synonymous (dN) mutation rates for codons in the HIF1α ODD domain (Fig. 3b). Interestingly, dN/dS ratios were lower in the vertebrate lineage than in the invertebrate lineage for most codons but the trend was most apparent for codons corresponding to the NODD and CODD regions (Fig. 3b). This suggests that the ODD domain in vertebrate species is under more stringent negative selection. Among the codons that had a lower dN/dS ratio in vertebrate species is Met$_{n-3}$. When considering the alignment of amino acid sequences

from 29 vertebrate and 22 invertebrate species, we observed that Met$_{n-3}$ is conserved in vertebrate species but variable in invertebrate species (Fig. 3c, Supplementary Fig. 3). Like HIF1α Met$_{n-3}$, HIF2α Thr$_{n-3}$ is conserved in vertebrate species. The one notable exception is the lamprey (P. marinus), where both HIF1α and HIF2α sequences contain Met$_{n-3}$ (Supplementary Fig. 3). Thus, not only are HIF1α Met$_{n-3}$ and HIF2α Thr$_{n-3}$ associated with distinct affinities for VHL, these residues are incredibly conserved in species expressing both HIF1α and HIF2α, suggesting that the specialization of VHL affinity is of importance.

**Methionine oxidation decreases HIF1α affinity for VHL**. Co-crystallization of HIF1αOH peptide with VBC complex has shown that HIF1α Met561 packs against VHL Phe91[10,11]. We recently co-crystallized HIF2αOH peptide with VBC complex and showed that substitution of Met$_{n-3}$ with Thr does not result in local conformational changes, with HIF2α Thr528 remaining in proximity to VHL Phe91[12]. It has been previously noted that the methionine-aromatic interaction is enriched within known protein structures and, according to quantum mechanical calculations, the interaction is predicted to contribute 1.0–1.5 kcal/mol of additional stability when compared to the interaction between an aromatic residue and a bulky nonpolar residue[27]. The average distance between the sulfur (Met) and aromatic ring center is 5 Å with a preferential orientation of 30–60°[27]. When analyzing two available HIF1α-VBC structures, the distance between the sulfur atom of HIF1α Met561 and the ring center of VHL Phe91 ranges between 4.58 Å (1LQB) and 4.80 Å (1LM8; Fig. 4a). The angle between the sulfur atom of HIF1α Met561 and the vector normal to the aromatic plane ranges from 41.3° to 42.9° (Fig. 4a). Concordantly, in silico analysis of the binding interface between VHL and HIF1α (1LM8) or VHL and HIF2α (6BVB) suggests that HIF1α Met561 contributes greater stability to the HIF/VHL complex than HIF2α Thr528 (1.41 kcal/mol versus 0.75 kcal/mol). Further, VHL Phe91 is more buried when bound by HIF1α peptide (41.2% buried surface area/accessible surface area) than by HIF2α peptide (32.7%). Thus, there appears to be a structural basis for the preferential binding of Met$_{n-3}$ containing peptides to VHL when compared to peptides where Met$_{n-3}$ is instead substituted with Thr or Ala.

Oxidation of methionine to methionine sulfoxide(MetO) has been implicated in cellular signalling. However, there is an ongoing discussion about whether oxidation of Met increases or

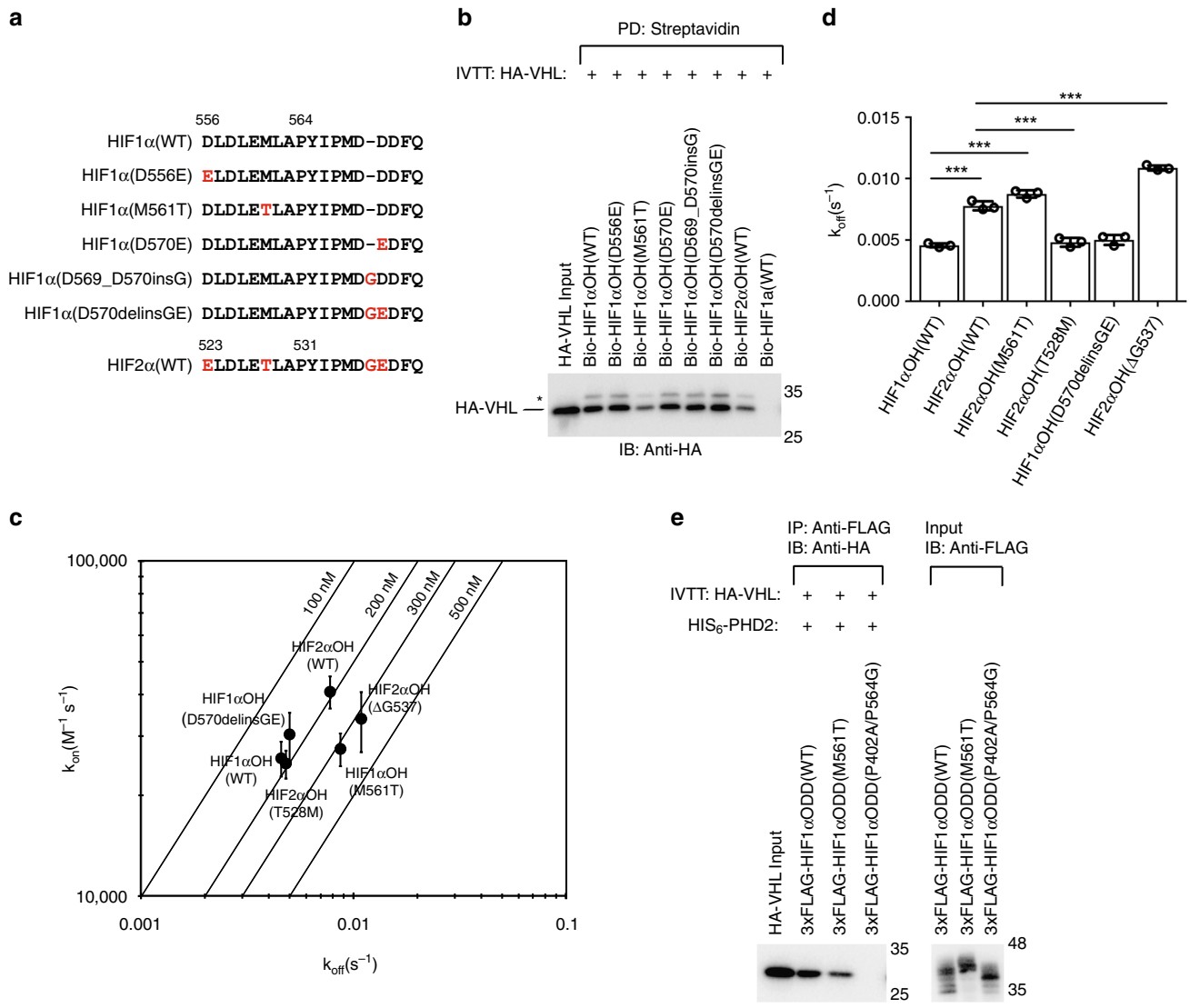

**Fig. 2** HIF1α Met$_{n-3}$ stabilizes interaction with VHL. **a** Peptide sequences used in subsequent experiments are listed. Peptides are N-terminal biotinylated. Numbering is provided for the N-terminal amino acid residue and the proline hydroxylation site. **b** Biotinylated HIFαOH peptides were immobilized on streptavidin- agarose beads and incubated with in vitro transcribed and translated (IVTT) HA-VHL. Streptavidin beads were pulled down (PD) and levels of HA-VHL were visualized via immunoblotting (IB). **c, d** Biolayer interferometry kinetic analysis of VHL-elongin B-elongin C (VBC) complex binding to immobilized HIFα peptides. Biotinylated peptides were immobilized on streptavidin- coated biosensors and binding to VBC complex was monitored. The data was analyzed assuming a 1:1 binding model via the BLItz Pro software. **c** Rate plane with Isoaffinity Diagonals (RaPID) plot highlighting the kinetic parameters of VBC complex binding to HIFα peptides. Values represent mean of three experiments conducted with independently purified protein ± s.e.m. **d** The dissociation constants associated with VBC binding to HIFα peptides are shown on a linear scale. Statistical significance was assessed using a one-way ANOVA with Tukey's post hoc test. Values represent mean of three experiments conducted with independently purified protein ± s.d. *** $p < 0.005$. **e** 3xFLAG-HIF1α oxygen-dependent degradation (ODD) domains were IVTT and incubated with purified HIS$_6$-PHD2 (181–426). Following hydroxylation (one hour), 3xFLAG-HIF1α ODD domain was immobilized on protein A beads coated with anti-FLAG antibody and incubated with IVTT HA-VHL. 3xFLAG-HIF1α ODD domain was immunoprecipitated (IP) and levels of HA-tagged VHL were visualized via immunoblotting (IB). Molecular weight markers (kDa) are labeled for western blots

decreases its affinity for aromatic residues. When the Met-aromatic interaction is simulated in the gaseous phase, it is evident that methionine oxidation increases the affinity of methionine for aromatic compounds, such as benzene[28,29]. However, the stability of the MetO-aromatic interaction is diminished when simulated in an aqueous environment. In an article by Lewis et al., the MetO-aromatic interaction was reported to still be more favourable than Met-aromatic in an aqueous environment yet a report by Orabi et al., instead suggested that Met interacts more strongly than MetO with aromatic compounds in an aqueous environment due to a

solvation effect driven by increased hydrophilicity[28,29]. As the tested HIFα peptides are poorly soluble, they are reconstituted in DMSO for most of the reported experiments. To ensure that the increased binding of HIF1α to VHL is not driven by potential artefactual oxidation of HIF1α Met561 to MetO561 in DMSO, we tested peptides dissolved in an aqueous buffer. HIF1αOH peptide solubilized in DMSO or an aqueous buffer pulled down similarly more VHL than the corresponding HIF2αOH peptides (Fig. 4b). However, synthetic oxidation of HIF1α Met561 but not Met568 resulted in decreased affinity for VHL (Fig. 4c). Complete oxidation of HIF1α Met561 to methionine sulfone (MetO$_2$)

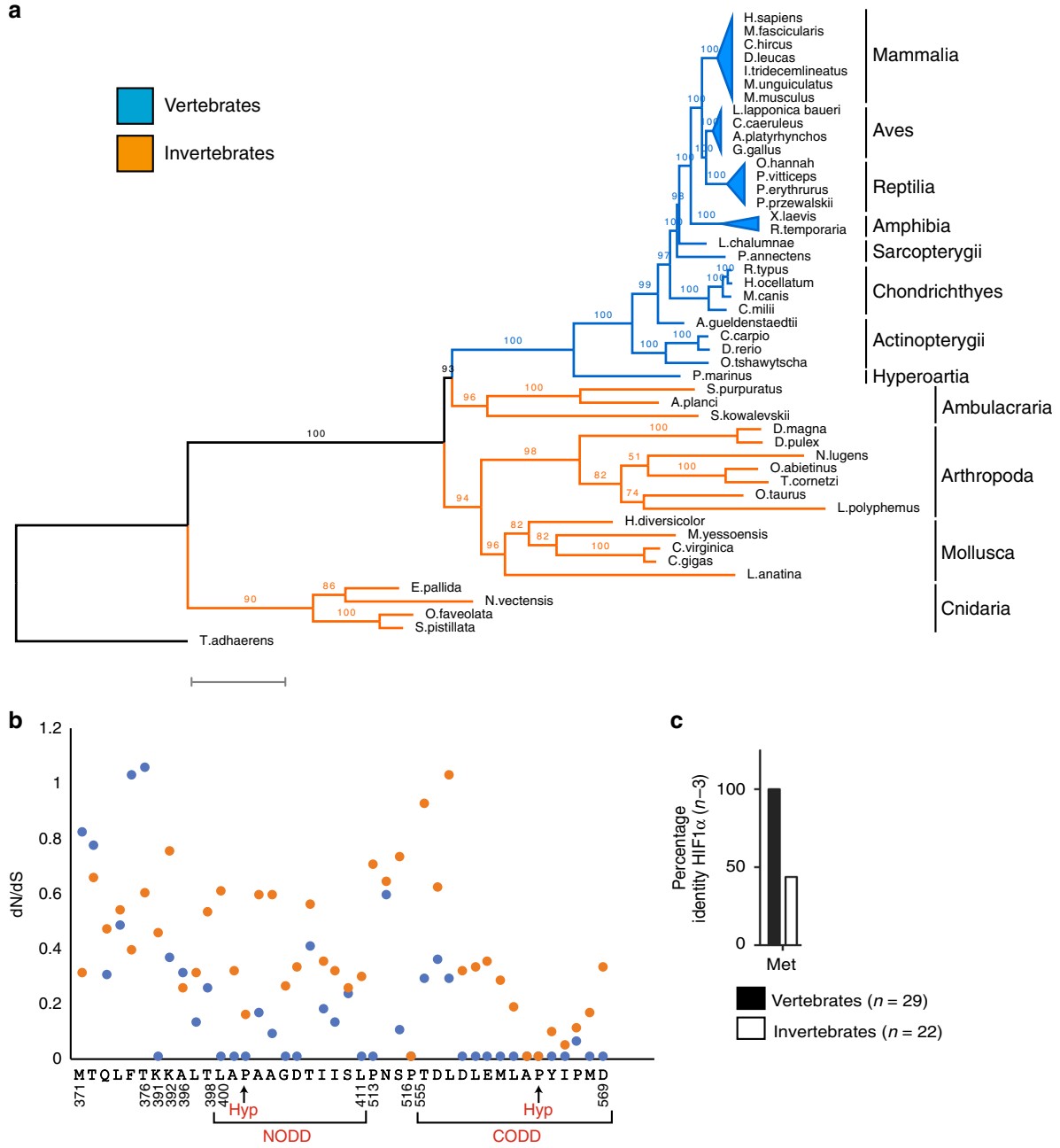

**Fig. 3** HIFα $Met_{n-3}$ is conserved in the vertebrate lineage. **a**, **b** Blue = vertebrate; orange = invertebrate. Full-length HIF1α cDNA sequences were aligned. The region corresponding to the oxygen-dependent degradation (ODD) domain was re-aligned and the top 21 alignments were concatenated. **a** A maximum-likelihood phylogenetic tree was constructed. Branch values indicate bootstrap support (100 bootstraps). Scale bar, 0.7 substitutions per site. **b** Non-synonymous (dN) and synonymous (dS) substitution rates were estimated for codons that aligned without gaps using the HyPhy software accessed via Mega7. The amino acid residue and numbering correspond to human HIF1α. **c** Annotated HIF1α amino acid sequences (full-length) were identified through use of BLASTP. Sequences were aligned using the MAFFT algorithm via the GUIDANCE2 webserver. The frequency of the amino acid three residues N-terminal of the primary hydroxylation site is indicated for vertebrate and invertebrate species

resulted in a further decrease in VHL affinity (Fig. 4c). Although oxidation of Met561 and Met568 resulted in a significant increase in the dissociation rate of VHL from HIF1αOH peptide, only oxidation of Met561 resulted in an increase in dissociation rate comparable to the M561T substitution (Fig. 4d, e). Substitution of HIF1α Met561 with norleucine did not affect the affinity of HIF1αOH peptide for VHL (Fig. 4c). These results suggest that the sulfur atom of methionine is not necessary for the increased stability of the HIF1αOH-VHL complex and that oxidation of HIF1α Met561 results in a decreased affinity for VHL, consistent with the model proposed by Orabi et al.

**Ancestral VHL protein binds poorly to HIFα.** Differential affinity of Met and Thr residues for aromatic residues provides a structural and biochemical rationale for differential binding of HIF1α and HIF2α to VHL. As we observed that HIF1α $Met_{n-3}$ is conserved in vertebrates but variable among invertebrate species, we asked whether VHL Phe91 followed a similar pattern of evolution. As the numbering of amino acids differs among species, we will refer to residues that align with VHL Phe91 as VHL $X_{n+3}$, where n refers to an invariant tryptophan residue (VHL Trp88 in humans) that forms part of the hydroxylproline binding pocket. Strikingly, we found that VHL $Phe_{n+3}$ is conserved in

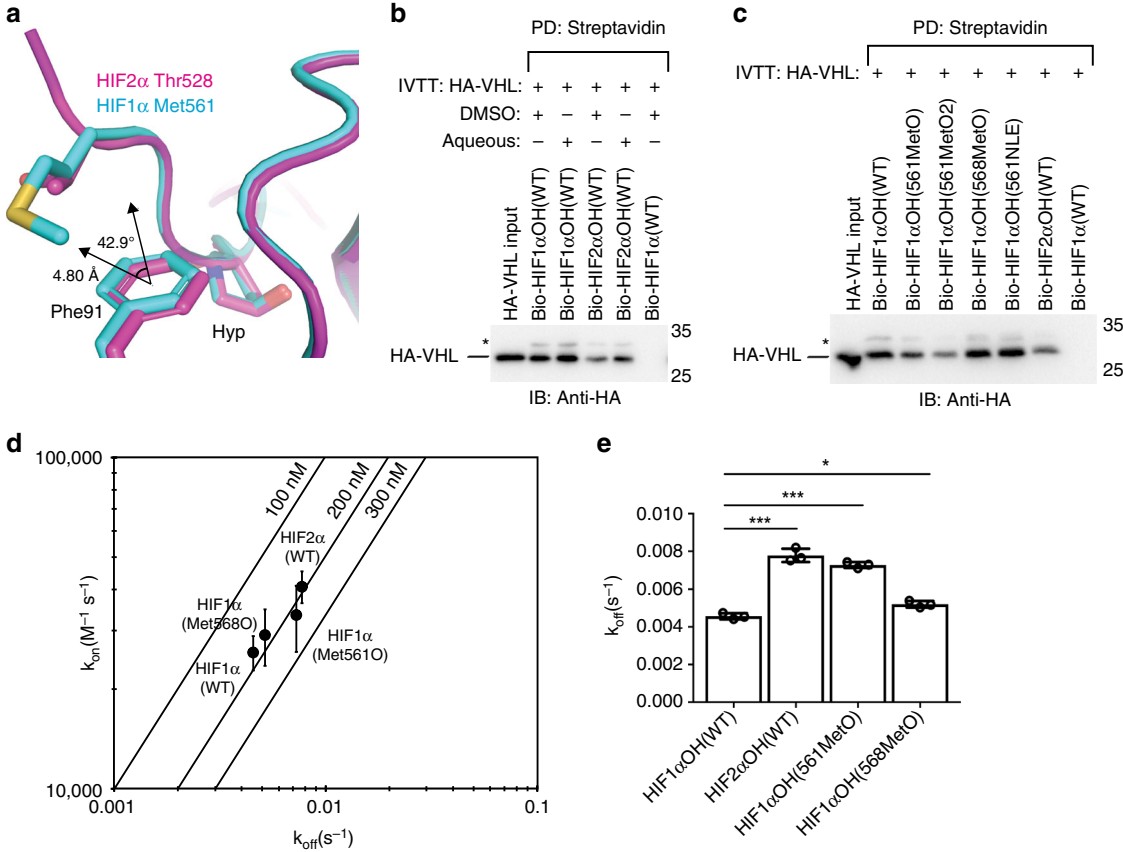

**Fig. 4** Oxidation of HIFα Met$_{n-3}$ weakens the interaction between VHL and HIFα. **a** Structure of VHL-HIF2α (6BVB) and VHL-HIF1α (1LM8) were accessed from PDB and superimposed using Pymol. The distance and angle between HIF1α Met561 (sulfur) and VHL Phe91 (center of aromatic ring) are indicated. **b**, **c** Biotinylated HIFαOH peptides were immobilized on streptavidin- agarose beads and incubated with in vitro transcribed and translated (IVTT) HA-VHL. Streptavidin beads were pulled down (PD) and levels of HA-VHL were visualized via immunoblotting (IB). **b** The solvent used to re-constitute the peptide is indicated. Molecular weight markers (kDa) are labeled for western blots. **d**, **e** Biolayer interferometry kinetic analysis of VHL-elongin B-elongin C (VBC) complex binding to immobilized HIFα peptides. Biotinylated peptides were immobilized on streptavidin- coated biosensors and binding to VBC complex was monitored. The data was analyzed assuming a 1:1 binding model via the BLItz Pro software. **d** Rate plane with Isoaffinity Diagonals (RaPID) plot highlighting the kinetic parameters of VBC complex binding to HIFα peptides. Values represent mean of three experiments conducted with independently purified protein ± s.e.m. **e** The dissociation constants associated with VBC binding to HIFα peptides are shown on a linear scale. Statistical significance was assessed using a one-way ANOVA with Tukey's post hoc test. Values represent mean of three experiments conducted with independently purified protein ± s.d. * $p < 0.05$, *** $p < 0.005$

vertebrate species but variable among invertebrate species (Supplementary Fig. 3, 4a). Among the studied invertebrate species, nearly all possessed an aromatic residue, with VHL Tyr$_{n+3}$ being the most prevalent (Supplementary Fig. 4a). Only *C. elegans* possesses a non-aromatic residue, instead featuring a bulky proline residue (Supplementary Fig. 3). Employing the Consurf server, HIF1α and VHL amino acid residues were colour-coded in accordance to degree of conservation among the vertebrate and invertebrate lineage (Fig. 5a). Residues that form the hydroxylproline binding pocket, including VHL Trp88, VHL Tyr98, VHL Ile109, VHL Ser111, VHL His115, and VHL Trp117, are almost invariably conserved in both vertebrate and invertebrate species. Conversely, HIF1α Met$_{n-3}$ and VHL Phe$_{n+3}$, which are adjacent to the hydroxylproline binding pocket, are variable in the invertebrate lineage (Fig. 5a).

Due to the intense negative selection that VHL Phe$_{n+3}$ is under in the vertebrate lineage, we next explored the pattern of HIF1α and HIF2α binding among VHL proteins with Phe91 substitutions. VHL F91W bound similarly to HIF1αOH and HIF2αOH peptides, when compared to VHL WT (Fig. 5b). Interestingly, VHL F91Y appeared to retain affinity for HIF1αOH peptide but bound less HIF2αOH peptide than VHL WT (Fig. 5b).

A F91L VHL mutation, which has been reported to cause VHL disease and sporadic RCC[30,31], was found to abolish binding to both HIF1α or HIF2α, suggesting that an aromatic residue in the n + 3 position is important for protein folding and/or stability. Further, we observed that VHL F91Y bound less HIF1αODD WT and HIF1α ODD M561T than VHL WT, suggesting that VHL F91Y binds more weakly to HIFα independent of HIFα sequence determinants (Fig. 5c). The failure to observe decreased binding of VHL F91Y to HIF1α WT in an in vitro pulldown assay may be due to saturation (Fig. 5b). Consistent with this notion, we observed that when less HIF1αOH peptide was loaded on streptavidin agarose beads, the weaker affinity of HIF1αOH peptide to VHL F91Y, compared with VHL WT, was readily observed (Supplementary Fig. 5). To further quantify the effect of the F91Y substitution on VHL/HIFα complex stability, we purified recombinant VHL F91Y (Supplementary Fig. 6). Using BLI, we confirmed that the F91Y substitution increases the dissociation rate of HIF1αOH and HIF2αOH peptide from VHL by roughly two-fold (Fig. 5d, e). Importantly, these results suggest that during the evolution of metazoans, particularly in the invertebrate lineage, substitution of VHL X$_{n+3}$ can modulate VHL/HIFα complex stability independently of HIFα sequence determinants.

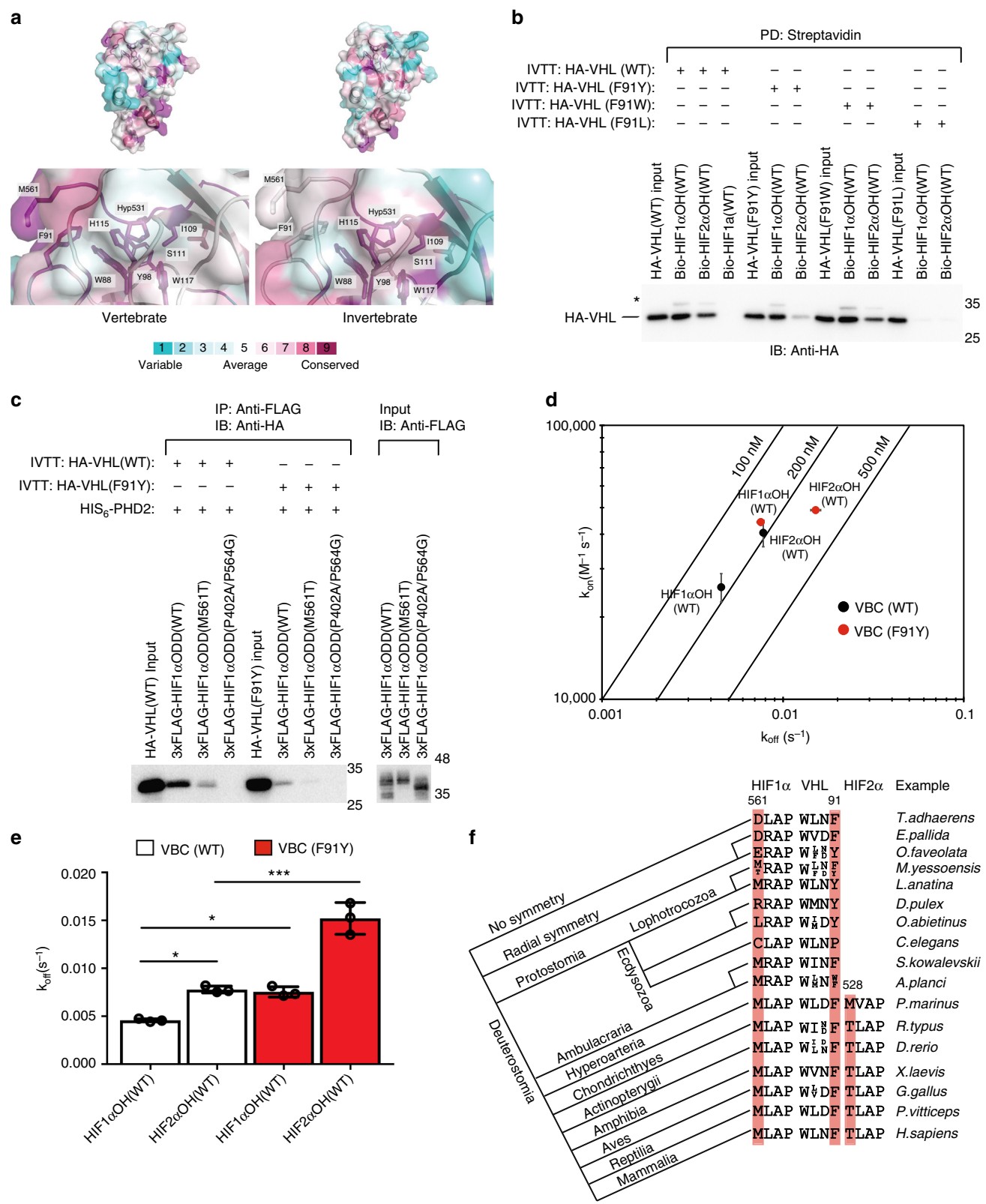

**Evolutionary fine-tuning of the HIF/VHL axis**. Having observed that HIFα Met$_{n-3}$ and VHL Phe$_{n+3}$ residues result in a more stable HIFα/VHL complex, we next asked when these motifs emerged during invertebrate evolution. First, we inferred ancestral HIF1α sequences using a maximum parsimony method (Supplementary Fig. 7). The analysis suggests that the last

common ancestor of radiata and bilateria possessed a HIFα featuring Glu$_{n-3}$ or Asp$_{n-3}$. However, the last common ancestor of deuterostome and protostome species is predicted to have featured HIFα Met$_{n-3}$. Among sampled protostome species, HIFα Met$_{n-3}$ is observed most commonly in lophotrochozoa while substitutions are frequently observed in ecdysozoa species. All

**Fig. 5** VHL Phe$_{n+3}$ is associated with a more stable VHL/HIFα complex. **a** The structure of HIF1α-VHL complex (PDB: 1LM8) is rendered as a cartoon superimposed with a transparent surface representation. Using multiple sequence alignments for VHL and HIFα in vertebrate and invertebrate species, amino acid residues were color-coded according to conservation using the Consurf webserver. **b** Biotinylated HIFαOH peptides were immobilized on streptavidin- agarose beads and incubated with in vitro transcribed and translated (IVTT) HA-VHL. Streptavidin beads were pulled down (PD) and levels of HA-VHL were visualized via immunoblotting (IB). **c** 3xFLAG-HIFα oxygen-dependent degradation (ODD) domains were IVTT and incubated with purified HIS$_6$-PHD2 (181–426). Following hydroxylation (one hour), 3xFLAG-HIFα ODD domain was immobilized on protein A beads coated with anti-FLAG antibody and incubated with IVTT HA-VHL. 3xFLAG-HIFα ODD domain was immunoprecipitated (IP) and levels of HA-tagged VHL were visualized via immunoblotting (IB). Molecular weight markers (kDa) are labeled for western blots. **d, e** Biolayer interferometry kinetic analysis of VHL-elongin B-elongin C (VBC) complex binding to immobilized HIFα peptides. Biotinylated peptides were immobilized on streptavidin- coated biosensors and binding to VBC complex was monitored. The data was analyzed assuming a 1:1 binding model via the BLItz Pro software. **d** Rate plane with Isoaffinity Diagonals (RaPID) plot highlighting the kinetic parameters of VBC complex, either WT of F91Y, binding to HIFα peptides. Values represent mean of three experiments conducted with independently purified protein ± s.e.m. **e** The dissociation constants associated with VBC binding to HIFα peptides are shown on a linear scale. Statistical significance was assessed using a one-way ANOVA with Tukey's post hoc test. Values represent mean of three experiments conducted with independently purified protein ± s.d. * $p < 0.05$, *** $p < 0.005$. **f** An idealized phylogenetic tree of metazoan evolution is depicted. Short HIF1α, HIF2α, and VHL sequences from representative species are given

chordata species sampled featured HIFα Met$_{n-3}$. We attempted a similar analysis with VHL sequences, but generation of a phylogenetic tree that accurately separated species into known clades proved difficult. Manual inspection of VHL sequences revealed that non-bilaterian species may possess either VHL Phe$_{n+3}$ or Tyr$_{n+3}$. Among protostome species, half of the examined lophotrochozoa possess VHL Phe$_{n+3}$, while all ecdysozoa species examined possess VHL Tyr$_{n+3}$. We next explored how often the VHL Phe$_{n+3}$ and HIFα Met$_{n-3}$ motifs co-exist in invertebrate species. As discussed, these residues are conserved in the vertebrate lineage. In the invertebrate lineage, more than half of all species (4/6) that contain HIF1α Met$_{n-3}$ possess a corresponding VHL Phe$_{n+3}$ residue (Supplementary Fig. 4b). Conversely, species with VHL Tyr$_{n+3}$ almost exclusively possess HIFα non-Met$_{n-3}$ (7/8). The particular residue varies depending on the clade; Leu in order hymenoptera (*T. cornetzi*, *O. abietinus*), Glu in order scleractinia (*O. faveolata*, *S. pistillata*), Arg in genus daphnia (*D. pulex*, *D. magna*; Supplementary Fig. 3). Additionally, *M. yessoensis* (scallop) features both VHL Tyr$_{n+3}$ and HIFα Thr$_{n-3}$, which co-immunoprecipitation and BLI studies revealed was the least stable HIFα-VHL complex, while other members of the family Ostreidae (*C. virginica*, *C. gigas*) possess both VHL Phe$_{n+3}$ and HIF1α Met$_{n-3}$, the most stable HIFα-VHL complex (Fig. 5c–e). These observations suggested that even within a group of closely related species, HIFα-VHL complex stability is variable and can potentially drive adaptation to their unique environments. Further, these observations led us to ask whether the combination of VHL Phe$_{n+3}$ and HIF1α Met$_{n-3}$ emerged once during evolution or along multiple lineages. To tackle this question, we constructed an idealized phylogeny on the basis of well-characterized branching points during metazoan evolution (radial symmetry versus bilateral symmetry, protostome versus deuterostome; Fig. 5f). Among ecdysozoa species for which we have VHL and HIFα sequence data, none possess the combination of HIFα Met$_{n-3}$ and VHL Phe$_{n+3}$. Among sampled lophotrochozoa, 2 out of 4 possess the combination of HIFα Met$_{n-3}$ and VHL Phe$_{n+3}$. With the ubiquity of the HIFα Met$_{n-3}$/VHL Phe$_{n+3}$ combination in deuterostomes and its absence in non-bilaterians, it becomes unclear whether species belonging to the ecdysozoa monophyletic clade experienced a loss of the HIFα Met$_{n-3}$/VHL Phe$_{n+3}$ combination or whether the presence of this combination in species belong to the lophotrochozoa clade represents independent, convergent evolution. Considering our ancestral sequence analysis, which suggested that HIFα Met$_{n-3}$ was present in the last common ancestor to lophotrochozoa and ecdysozoa species, a secondary loss in ecdysozoa seems more likely. However, a comprehensive phylogenetic study would need to be conducted in order to test this hypothesis. What is clear is that

among radiata and protostome species significant variation can be tolerated at the HIFα X$_{n-3}$ and VHL Phe$_{n+3}$ residues, while intense selective pressure seems to maintain HIFα Met$_{n-3}$ and VHL Phe$_{n+3}$ within the deuterostome clade.

In lamprey (*P. marinus*), a jawless, primitive vertebrate, both HIF1α and HIF2α feature Met$_{n-3}$ (Fig. 5f). This finding suggests that the ancestral HIFα that duplicated during the evolution of vertebrates possessed Met$_{n-3}$ and that Thr$_{n-3}$ emerged following divergence of the evolutionary lineage that led to modern-day lampreys, which occurred approximately 500 million years ago[32]. Although we are greatly interested in the evolutionary underpinning of HIF1α and HIF2α regulation in mammals, further studies would be required to elucidate the relative stabilities of ancestral HIFα/VHL complexes. Such studies may reveal insight into the adaptation of the oxygen/hypoxia-sensing pathway to various environmental conditions (e.g., terrestrial versus aquatic environments). Our evaluation of HIF1α M561A and NLE-substituted HIF1α suggests that substitution of Met$_{n-3}$ with less bulky amino acids would result in decreased complex stability. Further, we show that substitution of VHL Phe$_{n+3}$ with Tyr decreases HIFα/VHL complex stability irrespective of the HIFα X$_{n-3}$. Thus, substitution of these interacting residues may allow for a fine-tuning of the hypoxia response pathway via modulation of HIFα/VHL complex stability without abrogating VHL-mediated regulation of HIFα entirely.

## Discussion

Interaction of HIFα and VHL absolutely requires the hydroxylation of a conserved HIFα proline residue, which is hydrogen bonded to VHL residues and buried in a highly complementary pocket lined by bulky nonpolar residues. Here, we identify additional HIFα and VHL residues of evolutionary significance that can further stabilize the HIFα/VHL complex. The residues that line the hydroxylproline binding pocket are conserved in all examined metazoan species, suggesting that the mode of VHL/HIFα binding is well-preserved among invertebrate and vertebrate species (Fig. 5a). However, adjacent residues (i.e. HIFα X$_{n-3}$ and VHL X$_{n+3}$) can also interact and, depending on the biochemical properties of the residues substituted at those positions, can stabilize the VHL/HIFα complex to varying degrees. Among radiata, we predict, on the basis of sequence alignments (Supplementary Fig. 3), that VHL Tyr$_{n+3}$ is most prevalent and contacts a variable, non-Met residue on HIFα (Fig. 5f). Within the protostome clade, the combination of HIFα Met$_{n-3}$/VHL Phe$_{n+3}$ is observed in half of the sampled lophotrochozoa species but never in sampled ecdysozoa species (Fig. 5f). Conversely, among deuterostome species we instead observe that VHL Phe$_{n+3}$ and HIFα Met$_{n-3}$ are conserved (Fig. 5f). Our biochemical and

biophysical experiments would suggest that HIFα/VHL complex stability is increased in deuterostome invertebrates, from which vertebrate species evolved, compared to radiata and protostome species (Fig. 5c–e).

Two other evolutionary trends in hypoxia-sensing have been noted in invertebrate species. First, it has been well-described that HIFα in non-bilaterian species only have one ODD site while most protostome and deuterostome species feature two ODD sites. Second, while the *T. adhaerens* genome only encodes one PHD protein (most similar to human PHD2), invertebrate eumetazoan genomes, with the exception of protostomes, contain two PHD genes[7]. These two evolutionary trends have intersected. For example, human PHD3 specifically hydroxylates HIF1α CODD but not NODD[33]. Both of these evolutionary events (ODD and PHD duplication) appeared in the metazoan lineage before the HIFα $Met_{n-3}$/VHL $Phe_{n+3}$ secondary contact site became fixed. The evolution of a secondary ODD site and duplication of an ancestral PHD gene allows for a more fine-tuned, and possibly stronger, negative regulation.

Upon the evolution of vertebrate species, the ancestral HIFα gene went through several rounds of duplication, yielding multiple related transcription factors, with HIF1α and HIF2α thought to be the most important. Interestingly, an incredibly conserved substitution of HIF1α $Met_{n-3}$ to HIF2α $Thr_{n-3}$ occurred sometime after lamprey diverged from other vertebrate species (Fig. 5f). This substitution has been demonstrated to decrease the affinity of HIF2α for VHL by roughly two to three-fold when biochemical experiments are conducted with peptides and proteins corresponding to the human sequences (Fig. 1, Fig. 2, Supplementary Fig. 1, Supplementary Fig. 2). Thus, two important transitions need to be discussed; the transition from what is predicted to be a weaker HIF1α/VHL complex in some radiata and protostome species to a stronger complex in deuterostomes and the divergence of HIF1α and HIF2α, with respect to affinity for VHL, in the vertebrate lineage.

Animals first diversified between 600 and 500 million years ago under conditions that today would be described as hypoxic. One emergent model of hypoxia sensing and metazoan evolution focuses on the propensity for oxic niches to promote cell differentiation[34]. For example, primary non-malignant breast tissue can be maintained in an immature state when grown under hypoxic conditions, whereas normoxic conditions promote terminal differentiation[35]. An established body of work shows that hypoxia promotes survival of primitive hematopoietic stem cells[36–39]. Furthermore, activation of HIF2α is associated with sympathetic nervous system tumours with particular importance in the cancer stem cell population[40–42]. With these observations in mind, it has been suggested that the ability of HIF2α to promote a pseudohypoxic state conducive to self-renewal and proliferative properties in adult stem cell populations was evolutionarily advantageous[34]. By extension, rather than HIFα evolving as a mechanism to allow cells to survive low oxygen states, HIFα might have been necessary for the maintenance of stem cell populations under high oxygen conditions. The observation that HIF2α is stabilized under higher oxygen tensions (up to 7% $O_2$) than HIF1α (~2% $O_2$) suggests that, upon duplication of ancestral HIFα, the two paralogs specialized with HIF2α assuming the role of pseudohypoxia maintenance in select cell types[34]. However, in invertebrate species possessing only a single HIFα gene, a balance would need to be struck between the ability to maintain a pseudohypoxic niche and inducibility under oxygen tensions that impede on an organism's preferred metabolism. One can imagine that if pseudohypoxic maintenance was the only concern, an 'ideal' HIFα would lack an ODD domain entirely. We will now directly discuss our presented results with emphasis on this 'pseudohypoxic' model. We speculate that the relative rarity

of the HIFα $Met_{n-3}$/VHL $Phe_{n+3}$ motif in radiata and protostome species may represent an evolutionary pressure towards stabilization of HIFα at higher oxygen tensions (i.e. less efficient degradation). In this model, ancestral HIFα would be more stable and promote a pseudohypoxic state that is critical for stem cell niches yet could be further induced under hypoxia. Conversely, invertebrate deuterostome species possess HIF1α $Met_{n-3}$ and VHL $Phe_{n+3}$. In these species, other means of maintaining a pseudohypoxic niche may augment the HIF-mediated strategy. Generally speaking, the increased variability in HIF1α and VHL sequences among invertebrate species, particularly among the residues that we have studied, may reflect a fine-tuning of pseudohypoxic maintenance and metabolic regulation under hypoxic insult (Fig. 3b). If pseudohypoxic maintenance is an important HIFα role in invertebrate species, we would expect that HIFα protein should be present at detectable levels under ambient atmospheric conditions, much like HIF2α in mammalian cells.

One limitation of our interpretation of invertebrate HIF evolution is that we studied 'ancestral' HIF and VHL substitutions in the context of otherwise human protein and peptide sequences. Studies with full-length VHL and HIFα proteins from representative invertebrate species would be required to further speculate on the evolution of oxygen-sensitivity in HIFα proteins. For example, although radiata and protostome species rarely feature HIFα $Met_{n-3}$, HIFα $Arg_{n-3}$ and $Arg/Lys_{n-2}$ are frequently observed. Interestingly, Arg/Lys can participate in stable cation-π interactions with aromatic residues, which may functionally compensate for absence of HIFα $Met_{n-3}$[43]. As full-length HIFα proteins have been difficult to purify, another potential limitation of our study is the use of HIFα peptides as a surrogate for full-length or ODD recombinant protein. However, the peptide design we utilized is sufficient for VHL interaction, and the affinity of IVTT HIFα ODD protein for VHL can be modulated by substitution of amino acid $X_{n-3}$ in a manner that mirrors our biophysical and biochemical peptide studies (Fig. 2f, Supplementary Fig. 1c). As our immunoprecipitation studies with HIFα ODD and VHL require prior hydroxylation of HIFα via recombinant PHD2, we have reason to believe that overall post-translational regulation of HIF1α is weakened by the $Thr_{n-3}$ substitution. However, it remains important to study how PHD-mediated regulation of HIFα has evolved in metazoans, paying close attention to the variable number of HIFα ODD sites and PHD proteins expressed in various invertebrate species.

Early vertebrate genome duplications also contributed to the diversification and specialization of HIF pathway as an expanded repertoire of hypoxia sensitive transcription factors allowed for functional specialization. Our biophysical study of HIF1α and HIF2α binding affinity for VHL suggests that substitutions of amino acid residues proximal to the primary hydroxylation site played a role in the divergence of HIF1α and HIF2α in the vertebrate lineage. Although research into the biochemistry of HIFα in invertebrate species is lacking, evolution of the HIF1α coding sequence has been studied extensively in fish, including the family cyprinidae, which retained two copies of HIF1α (HIF1αA/B) and HIF2α (HIF2αA/B). In cyprinids, NODD is lost specifically in HIF1αA[44,45]. In vivaporous eelpout (*Zoarces viviparous*), which belongs to the family Zoarcidae, the NODD proline is substituted with a leucine residue[46]. In teleost species, the NODD region is generally less conserved than the CODD region[46]. Moreover, human HIF2α NODD is poorly hydroxylated by PHD2, which can effectively hydroxylate HIF1α NODD[47]. These observations correlate well with the elevated rate of non-synonymous mutation in HIF2α NODD in vertebrates, including both teleosts and mammals, when compared to either HIF2α CODD or HIF1α NODD[26]. Generally, these substitutions appear to affect only one paralog, resulting in an increasingly fine-tuned hypoxia response

in vertebrate species, where one gene paralog is potentially stabilized over a wider range of oxygen tensions and may be of particular importance for pseudohypoxia maintenance.

Further work has been conducted to explore whether substitution of HIFα residues witnessed along an evolutionary branch can modulate HIFα stability. A more direct parallel to our observation is seen in Cyprinid fish belonging to the genus Schizothorax, which are well-adapted to hypoxic conditions present in the Tibetan plateau. These fish exhibit a substitution of HIF1αB $L_{n-5}$, which is associated with increased stability of the protein[44]. Other missense HIF2α mutations have been observed to be enriched in high-altitude populations, including a Q579L mutation in Tibetan cashmere goat (Capra hircus)[48] and G305S in several species of high-altitude dog[49]. However, the functional effects of these mutations, if any, have yet to be discerned. Overall, it is clear that substitutions in the coding sequence of HIFα allows for modulation of HIFα signalling activity, in part, by modulating negative regulation via PHD and/or VHL.

We observed that the VHL F91Y substitution, which is invariant in vertebrate species, resulted in lower affinity for both HIF2α and HIF1α (Fig. 5c–f). The intense purifying selection that maintains VHL $Phe_{n+3}$ in the vertebrate lineage may reflect a decrease in fitness associated with excessive stabilisation of HIF2α. Concordantly, HIF2α-activating mutations in humans have been associated with either polycythemia, a condition of elevated hematocrit, or neuroendocrine tumours, including paraganglioma, pheochromocytoma, and somatostatinoma[41,50]. We previously identified disease-causing mutations of residues proximal to the CODD hydroxylation site that result in only mild stabilisation of the HIF2α protein[12]. Thus, further stabilisation of HIF2α can be deleterious at the organismal level.

Although the strong conservation of HIFα $Met_{n-3}$ in the vertebrate lineage does support our hypothesis that divergence of HIF1α and HIF2α affinity for VHL is of biological importance, our VHL/HIFα binding experiments also suggest that the insertion of HIF2α $Gly_{n+6}$ does not overtly modulate complex stability (Fig. 2b, d). Thus, it appears that conservation of a residue proximal to HIFα hydroxylation sites does not necessarily suggest that the residue is of importance but instead may reflect fixation of a substitution that is (nearly) neutral. However, a second possibility is that HIF2α $Gly_{n+6}$ modulates regulation via PHD enzymes. It has been previously shown that modulation of the distance between HIF1α P564 and HIF1α L574, by insertion or deletion of aspartate residues, increases PHD-mediated hydroxylation[51]. Thus, it is possible that an insertion of glycine alters the interaction of HIF2α with PHDs.

In conclusion, we reveal that HIFα/VHL complex stability is modulated by substitutions in HIFα and VHL residues that are relevant to metazoan evolution. Notably, human HIF1α and HIF2α proteins intrinsically differ in their affinity for VHL and that the underlying molecular evolution suggests that specialization of HIF1α and HIF2α in the vertebrate lineage has been maintained by intense purifying selection. Future work on the molecular evolution of HIFα and VHL may reveal further insight into the unique roles of HIF1α and HIF2α in human biology and the dual roles of metabolic regulation and pseudohypoxic maintenance in early metazoan evolution.

## Methods

**Plasmids.** Construction of the following plasmids has been described previously; pcDNA3-HA-VHL$_{30}$, pcDNA3-HA-HIF1α[3], pACYCDuet-1 plasmid encoding untagged elongin B and elongin C(17–112), pGEX-4T-1-GST-VHL$_{19}$(54–213)[52], pcDNA3–3xFLAG-HIF2α ODD(390–554; both WT and P531A), and pET-46-HIS$_6$-PHD2[12]. F91Y, F91L, and F91W mutations were introduced into the pcDNA3-HA-VHL$_{30}$ and pGEX-4T-1-GST-VHL$_{19}$ plasmids via site-directed mutagenesis. pcDNA3–3xFLAG-HIF2α ODD P405A/P531A double-mutant was generated via site-directed mutagenesis of pcDNA3-3xFLAG-HIF2α ODD P531A.

pcDNA3-3xFLAG-HIF1α ODD (387–581) was sub-cloned from full-length pcDNA3-HA-HIF1α. The pcDNA3-3xFLAG-HIF1α ODD (387–581; P402A/P564G double-mutant) was sub-cloned from a full-length HIF1α P402A/P564G construct generously gifted by from Dr. Norma Masson and the lab of Dr. Peter Ratcliffe. HIF2α T528M and HIF1α M561T constructs were generated in both the ODD and full-length HIFα vectors. All primers used for cloning are listed in Supplementary Table 1.

**Antibodies.** Anti-HA (C29F4; 1:2000 dilution western blot) was obtained from Cell Signaling Technology. Anti-FLAG (F1804; 1:5000 dilution western blot; 1:1000 dilution immunoprecipitation) was obtained from Sigma-Aldrich.

**Peptides.** HIF-α peptides, with N-terminal biotinylation and C-terminal amidation modifications, were custom synthesized by Genscript. For pVHL interaction studies, WT HIF1α (556–573; DLDLEMLA[Hyp]YIPMDDDFQ) and WT HIF2α (523–541; ELDLETLA[Hyp]YIPMDGEDFQ) were used, where Hyp denotes hydroxylproline. Peptides were reconstituted to 2 mg/mL, as measured by A280, using either sterile DMSO or 0.1 M ammonium bicarbonate pH 9.0 buffer. Peptides were aliquoted and stored at −80 °C.

**In vitro binding assay.** The in vitro pVHL-HIFα binding assay was performed as previously described[52]. HA-VHL$_{30}$ was expressed in an IVTT rabbit reticulocyte lysate system (Promega, Cat. No. L1170) and incubated with 1.2 μg of biotinylated HIFαOH peptide immobilized on streptavidin agarose beads in 500 μL of EBC buffer (50 mM Tris-HCl pH 8.0, 120 mM NaCl, 0.5% (v/v) NP-40) for two hours at 4 °C. To perform a competition binding assay, 1.8 μg of biotinylated HIFαOH peptide immobilized on streptavidin agarose beads competed with 1 μg of acetylated HIFαOH peptide. Following incubation, beads were washed 4x with NETN buffer (20 mM Tris-HCl pH 8.0, 100 mM NaCl, 1 mM EDTA, 0.5% (v/v) NP-40). Biotinylated peptide was pulled down via streptavidin agarose beads and protein was eluted by boiling beads in sample buffer. Protein levels of HA-VHL$_{30}$ were determined by immunoblotting.

**Biolayer interferometry.** The binding affinities of VBC to hydroxylated HIFα peptide was measured by biolayer interferometry using the BLItz system (Pall ForteBio). A 50 μg/mL solution of biotinylated HIFα peptide was prepared in kinetics buffer (20 mM HEPES pH 7.4, 200 mM NaCl, 1 mM DTT, 0.02% (v/v) Tween-20, and 0.5% (w/v) BSA) and immobilized onto streptavidin (SA)-coated biolayer interferometry (BLI) biosensors (Pall ForteBio, Cat. No. 18–5019) over 120 s. Multiple concentrations of purified VBC complex were diluted in kinetics buffer and allowed to associate with immobilized peptide over 120 s. Subsequently, the SA-BLI probe was immersed into kinetics buffer for 120 s to allow for dissociation. Both protein and buffer were chilled on ice immediately before use. The data were analyzed, step corrected, reference corrected, and fit to a global 1:1 binding model. $K_d$, $k_a$, and $k_d$ were calculated using the BLItz Pro software. All measurements were performed in triplicate with independently purified recombinant protein.

**Surface plasmon resonance.** Single cycle kinetics experiments were conducted with the Biacore X100 (GE Healthcare Life Sciences, Cat. No. BR110073) at 25 °C. A 125 ng/mL solution of HIFα peptide was prepared in running buffer (20 mM HEPES pH 7.4, 200 mM NaCl, 0.02% (v/v) Tween-20, 1 mg/mL BSA) and immobilized using a biotin CAPture kit (GE Healthcare Life Sciences, Cat. No. 28920233) over 120 s. VBC was serially diluted in running buffer (0.123 μg/mL, 0.37 μg/mL, 1.11 μg/mL, 3.33 μg/mL, 10 μg/mL) and flowed over the SPR chip for 120 s at a flow rate of 30 μL/min, with a final dissociation step of 600 s. The data was double-referenced and fit to a global 1:1 binding model.

**In vitro hydroxylation assay.** 3xFLAG-HIFα ODD was expressed via the rabbit reticulocyte lysate system. HIFα ODD was incubated with 15 μg/μL HIS$_6$-PHD2 (181–426) in 200 μL of 40 mM HEPES pH 7.4, 80 mM KCl, 5 mM a-ketoglutarate, 2 mM ascorbic acid, 100 μM FeCl$_2$ tetrahydrate solution at 30 °C for 1 h. Following hydroxylation, 3xFLAG-HIFα was incubated with IVTT VHL$_{30}$ in 500 μL of wash buffer (40 mM HEPES pH 7.4, 80 mM KCl, 0.05% (v/v) Tween-20, 250 μM EDTA, 1 mM DTT) supplemented with protease inhibitors for 2 h at 4 °C. Following binding, 3xFLAG-HIFα ODD was immunoprecipitated with anti-FLAG antibody immobilized on protein A beads. Protein was eluted by boiling beads in sample buffer. Protein levels of HA-VHL$_{30}$ were determined by immunoblotting.

**Structural analysis.** Structures of VHL-HIF1αOH complex (1LM8, 1LQB)[10,11] and VHL-HIF2αOH complex (6BVB)[12] were accessed via PDB. Structures were superimposed using Pymol. Proteins, Interfaces, Structures, and Assemblies was used to evaluate the structural interface of HIFα/VHL.

**Alignment.** Annotated HIF1α, HIF2α, and VHL sequences were identified with the BLASTP algorithm accessed with the Mega7 software[53]. Sequence identifiers are listed in Supplementary Table 2. Full-length amino acid sequences were aligned

using the GUIDANCE2 webserver via the MAFFT algorithm with 100 bootstrap repeats and 1000 cycles of iterative refinement[54]. Full-length HIF1α cDNA sequences were aligned as codons using the GUIDANCE2 webserver as described above. Nematode sequences were not included for multiple sequence alignment (MSA) of cDNA as they were unreliably aligned according to confidence scores generated by GUIDANCE2. The ODD domain was defined as the sequence that aligned with human HIF1α codons 359–598. ODD sequences were re-aligned as described above. A SuperMSA was created by concatenating the top alignment with 20 alternative alignments. The SuperMSA contained 10435 codons, including gaps. Alignments were visualized using ESPript3[55].

**Phylogenetic analysis.** The SuperMSA was used to generate a reference phylogenetic tree. Smart Model Selection (SMS) was accessed via the PhyML 3.0 webserver to select a model for generating a maximum-likelihood phylogenetic tree[56]. A phylogenetic tree was constructed with a generalized time reversible (GTR) substitution model with four substitution rate categories (gamma factor of 1.374) and a fraction of sites allowed to be invariant (0.015)[57]. Branch support was determined by performing 100 bootstraps. *T. adhaerens* was selected as the outgroup. FigTree was used to visualize phylogenetic trees (http://tree.bio.ed.ac.uk/software/figtree/).

**Estimating selective pressure.** The HyPhy package, accessed via Mega7 software, was used to estimate synonymous and non-synonymous mutation rates for codons among vertebrates and invertebrate species based on the top MSA generated via GUIDANCE2. The above-described maximum-likelihood phylogenetic tree was used as reference. The GTR substitution model was used and all columns containing gaps were deleted.

Consurf was used to generate visual representations of HIF1α and VHL evolutionary conservation at the amino acid level[58]. The co-structure of HIF1α peptide bound to VHL was accessed from PDB (1LM8). Separate amino acid alignment of VHL and HIF1α were provided for vertebrate and invertebrate species. The *H. sapien* sequence was included for the invertebrate alignment to accurately project alignment information onto the structure of human VHL bound to HIF1α.

**Inferring ancestral sequence.** Using Mega7 software, ancestral amino acid states were inferred using the Maximum Parsimony method. The set of states at each node is ordered from most likely to least likely, excluding states with probabilities below 5%. The above-described maximum-likelihood phylogenetic tree was used as reference.

**Quantification and statistical analysis.** To evaluate statistical significance of dissociation constants, a one-way ANOVA with Tukey post hoc test was conducted using Prism software. A *p* value below 0.05 was considered significant.

**Reporting summary.** Further information on research design is available in the Nature Research Reporting Summary linked to this article.

## Data availability

Source data for Figs. 1a, 1b, 2b, 2c, 2d, 2e, 3a, 3b, 3c, 4b, 4c, 4d, 4e, 5b, 5c, 5d, 5e, and supplementary Figs. 1a, 1b, 1c, 2a, 2b, 2c, 4a, 4b, 5, 6b are provided as a Source Data file. All data are available from the corresponding author upon reasonable request.

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

## Acknowledgements

We thank the members of Ohh and Lee labs for their critical comments and discussions. We also thank Drs. Megha Shah, Trevor F. Moraes and James M. Rini for help with SPR experimentation and analysis. This work was supported by grants from the Canadian Institutes of Health Research (CIHR) (MOP-133694 to J.E.L.; PJT-159773 to M.O.). D.T. is a recipient of Vanier-CIHR Canada Graduate Scholarship.

## Author contributions

M.O. and D.T. conceptualized the project and study design. D.T. performed all experiments. J.E.L. aided in designing, performing, and writing the biophysical experiments. D.T. and M.O. wrote the paper.

## Additional information

**Competing interests:** The authors declare no competing interests.

