## [Peer Review File · Nature Communications]

Reviewers' comments:

Reviewer #1 (Remarks to the Author):

Tarade et al. describe an interesting study on the VHL-HIF-alpha interaction specificity and evolution. This is a pathway of enormous relevance, both physiologically and therapeutically, and its evolution is an area of growing interest, following seminal studies by Loenarz et al (2011, Embo Rep) and more recently Mills et al (2018, ELife).

The authors observe a key noncovalent interaction occurring in close proximity to the crucial hydroxyprolines (Hyp) that involves a Met residue in HIF-1alpha (corresponding to Leu in HIF-2alpha) at a position three residues upstream of Hyp (n-3) and a conserved Phe residue (Phe91) on VHL. This interaction leads to stronger binding of VHL to HIF-1 vs HIF-2, as shown by pull-down experiments and biolayer interferometry (BLI), and mutational studies support the important role played by this pair of residues. The authors then extend to a bioinformatics and evolutionary sequence analysis, and found that both residues (the n-3 residue on HIF, and the Phe91 residue in VHL) are strictly conserved in vertebrates but less conserved in invertebrates. These observations collectively support the author's analysis for how the HIF gene duplicated during evolution, and HIF-1 evolved into a higher-affinity binding to VHL as a mechanism to fine tuning oxygen sensing during evolution. Overall, this is a beautifully written manuscript, and a very nice study, and I congratulate the author on this. Overall, the data supports the paper's claim that metazoan evolution of the oxygen-sensing pathway involved conserved divergence of VHL affinity for HIF1 α and HIF2 α .

While I am supportive of publication in principle in the journal, a weakness of the current version is that the biophysical data presented is not of adequate quality to support the important conclusions on the relative differences in binding affinity and in particular dissociation kinetics. These differences are clearly subtle, for example 2-3 fold overall between HIF-1a and HIF-2a due to the Met-Leu change. So it is absolutely paramount that the best, most sensitive and highest quality of biophysical measurements is obtained. Unfortunately, the BLI data obtained is problematic. Visual inspection of the BLI data presented (Figures 2C and Figure S2) shows clearly that the curvatures of the fitting curves (both in the association and dissociation phases) depart significantly from the curvature of the observed data. This suggests that the simple 1:1 binding model is not adequate to fit the data, most likely due to mass transport limitation, re-binding or other artefactual phenomena to do with the technology. A simple plot of the residuals (difference between data and fit) as a function of time for each concentration curves will visually and clearly illustrate the problem. A model that adequately fits the data should yield residuals that are distributed randomly. Instead, when residuals are non-random, and instead follow a pattern, that strongly suggests that the model is not adequate to fit the data. Inspection of the data included in the paper suggests that the residuals (once shown) would prove to be non-random. This is perhaps not surprising given that the data was recorded with the Blitz instrument from ForteBio, which is really their entry instrument and best suited for quick checks of protein presence in crude matrices as well as at various stages of purification, and perhaps high affinity protein-protein interactions such as antibodies. Also it is widely accepted in the field that SPR is much superior than BLI to resolve kinetics. The authors are strongly encouraged to utilise an alternative biophysical technique to measure binding affinities and dissociation off-rates. I would strongly recommend SPR e.g. Biacore 3000 or even better T100/200, which would guarantee much more robust data and conclusions.

Other minor points:

- Figure 2C: confusing to show representative BLI data with HIF1alpha-OH and HIF-2alpha

(presumably Pro instead of Hyp) – as it could be misleadingly interpreted as HIF1alpha peptide binds while HIF2alpha peptide does not. To avoid this, the authors should show either data for the same HIF e.g. HIF-1, in one case Hyp in the other case Pro; alternatively, show data for both HIF-1alpha-OH and HIF-2alpha-OH

- “Although oxidation of Met561 and Met568 resulted in a significant increase in the dissociation rate of VHL from HIF1 α OH peptide, only oxidation of Met561 resulted in a decrease in dissociation rate comparable to the M561T substitution (Fig. 4d, e). “ please revise to ensure increase vs decrease of dissociation rate are described correctly

- The authors might want to streamline the Discussion to make it more concise and clear.

Reviewer #2 (Remarks to the Author):

This manuscript of Tarade et al report how the divergence hypoxia inducible factors HIF1alpha and HIF2alpha involved specific amino acid substitutions that affect their affinity to von Hippel-Lindau (VHL) tumor suppressor protein. This in turn partly may contribute to the observed difference of HIF1alpha and HIF2alpha in moderate oxygen tensions, HIF2alpha having weaker interaction with VHL and higher stability. Previously, HIF fragments containing the parts of the oxygen dependent domain including immediate surrounding of the critical proline have often been studied in the context of their interaction with prolyl hydroxylases that require oxygen as a substrate for the reaction. In this study the HIF fragments are used to study the HIF interaction with VHL that tags HIF to degradation after hydroxylation of the critical proline. The key novelties of the paper include the combination of phylogenetically informed selection of diverged amino acid positions for series of mutations for kinetic measurements, and the finding that the mutation of HIF1 α Metn-3 to “HIF2 α type” Thr has a significant effect on the affinity of VHL for the HIF peptide. This finding is of general interest for the field of hypoxia research, but also evolutionary biology. When considering the evolution of the three components of the HIF system: HIFs and PHDs have both 3 paralogs in mammals whereas VHL only one – this study highlights that the interactions with the non-duplicated component of the system has been important in the evolution of the complex as a whole. The methodology, presentation and reporting are convincing, and in key experiments three independent repeats were conducted. The figures and supplementary data are clear and the methods are described sufficiently. In the discussion part the argumentation is partly coupled to a recent conceptual model introduced by Hammarlund et al 2018, but the results of this paper have limited applicability to substantiate this model. However, overall, this paper advances our understanding of the evolution of HIF pathway and the divergence of HIF alpha paralogs and is a valuable scientific contribution.

Major revisions:

1. In the discussion the authors focus on how HIF CODD Metn-3 influences VHL interaction when looking at the evolution of HIF pathway. The authors should add a paragraph to discuss the new findings also in the light of relative contribution of HIF-VHL interactions versus HIF-PHD interactions in the evolution of HIF mediated oxygen-sensing. Also, the combined contribution of NODD and CODD to the VHL interaction could be touched on. Further, invertebrates encode a variable sets of HIF components in their genomes that based on their conservation are predicted to have functional importance. For example, roughly, placozoa has only CODD and one PHD, cnidarians have only CODD and two PHDs, and chordates both NODD and CODD as well as two PHDs. How does the variation in the discussed Metn-3 and VHL correlate with the presence of multiple ODD or PHDs?

2. Authors should comment on how the experiments using short HIF peptides (now 19 amino acids) present a reasonable approximation of the whole HIF CODD region or the HIF protein interaction with VHL.

3. Design for Figures 3c, 5a, 6a and 6b. Showing variable positions of a-a s with % bars is useful for the vertebrate vs invertebrate comparison, but not useful for looking at ancestral states among invertebrates. The figure with phylogeny is brought in maybe a bit too late in 6b, I suggest that 6b could be introduced earlier and in a more reader friendly format. The current % figs 5a 6a could be in the supplementary, and for the main figure(s) it might be better to modify 6b tree that focuses on invertebrates, so that you depict the pattern of crucial amino acids in HIF and VHL (and possible also the branch specific dn/ds) on the branches of the tree - not just alignment. Using some ancestral state reconstruction method (see Table 1 in Joy JB 2016 PLoS Comput Bio) would substantiate the subject that you present on page 6. " By constructing an idealized phylogeny on the basis of well-characterized branching points during metazoan evolution (radial symmetry versus bilateral symmetry, protostome versus deuterostome), several possible evolutionary trends are possible. The simplest model suggests that the ancestor to extant deuterostome species possessed a combination of VHL Phen+3 and HIF1 α Metn-3...."

4. Page 7. " Conversely, among deuterostome species we instead observe that VHL Phen+3 and HIF1 α Metn-3 are conserved (Fig. 6b). Our biochemical and biophysical experiments would suggest that HIF1 α /VHL complex stability is increased in deuterostome invertebrates, from which vertebrate species evolved, compared to radiata and protostome species (Fig. 5d, e, f). Upon the evolution of vertebrate species, the ancestral HIF1 α gene..."

- In invertebrates there is HIF α not HIF1 α .

5. Page 8. "One emergent model of hypoxia sensing and metazoan evolution focuses on the propensity for oxic niches to promote cell differentiation (29). ... By extension, rather than HIF α evolving as a mechanism to allow cells to survive low oxygen states, HIF α might have been necessary for the maintenance of stem cell populations under high oxygen conditions."

- Please, directly state that now the results will be discussed with emphasis on this (new) model. Also, if possible, directly state if the results presented in this paper support this model. The model has quite reasonable arguments, but it remains a bit unclear if it is associated with a testable hypothesis/predictions. Maybe it would be useful to add short description on if and how your results fit better the "pseudohypoxia model" rather than its simplistic "counterargument". "Counterargument" or old line of thought being (here just roughly simplified), that in invertebrate deuterostomes the increased affinity of VHL-HIF would present pressure towards tighter hypoxia dependent regulation of HIF and then in vertebrates the selection on HIF1 and HIF2 lead to wider range of stabilization in different oxygen tension to enhance more fine-tuned hypoxia responses.

Minor revisions:

Page 3. In addition to (17), add Smythies JA et al (2019) for chromatin binding

Page 3. "Despite well-documented differences between HIF1 α and HIF2 α , little is known about how processes of natural selection acted upon the coding region of these two genes to optimize the cellular response to changes in oxygen during evolution of the metazoans."

- do not make a paragraph out of one sentence.

- In invertebrates there is HIF α and in vertebrates HIF1 α and HIF2 α , thus instead of metazoans in this

sentence structure it should be vertebrates.

- Modify, for example mention that natural selection on these two genes has only been studied using computational sequence analysis (such as Rytönen KT et al 2011) but the role of specific amino acid substitutions have not been assessed by kinetic measurements of protein function...

Page 3.

"We have previously noted that in the immediate vicinity of the primary hydroxylation sites..."

- Add a reference

Page 4. ... Fig 3b, modify the figure so that the readers can immediately see when NODD changes to CODD, for example add a space,

Below the figure, for clarity, mark hydroxyprolines with red.

Page 8. "However, as noted above, studies with full-length VHL and HIF α proteins from representative invertebrate species would be required to further speculate on the evolution of oxygen-sensitivity in HIF α proteins."

- also consider that PHDs are important for the system, and invertebrates have variable numbers PHD gene copies and HIF ODDs.

Page 8. "One advantage of vertebrate hypoxia-sensing is the expanded repertoire of hypoxia sensitive transcription factors allowing specialization"

- This is an odd sentence, modify, for example: early vertebrate genome duplications also contributed to the diversification and specialization of HIF pathway

Page 9. "Furthermore, these substitutions appear to affect only one paralog, resulting in an increasingly fine-tuned hypoxia response in vertebrate species, where one species is stabilized over a wider range of oxygen tensions and may be particularly important for pseudohypoxia maintenance."

- species ? = gene paralog

RESPONSE TO REVIEWERS

REVIEWER No. 1

Tarade et al. describe an interesting study on the VHL-HIF-alpha interaction specificity and evolution. This is a pathway of enormous relevance, both physiologically and therapeutically, and its evolution is an area of growing interest, following seminal studies by Loenarz et al (2011, Embo Rep) and more recently Mills et al (2018, ELife).

The authors observe a key noncovalent interaction occurring in close proximity to the crucial hydroxyprolines (Hyp) that involves a Met residue in HIF-1alpha (corresponding to Leu in HIF-2alpha) at a position three residues upstream of Hyp (n-3) and a conserved Phe residue (Phe91) on VHL. This interaction leads to stronger binding of VHL to HIF-1 vs HIF-2, as shown by pull-down experiments and biolayer interferometry (BLI), and mutational studies support the important role played by this pair of residues. The authors then extend to a bioinformatics and evolutionary sequence analysis, and found that both residues (the n-3 residue on HIF, and the Phe91 residue in VHL) are strictly conserved in vertebrates but less conserved in invertebrates. These observations collectively support the author's analysis for how the HIF gene duplicated during evolution, and HIF-1 evolved into a higher-affinity binding to VHL as a mechanism to fine tuning oxygen sensing during evolution. Overall, this is a beautifully written manuscript, and a very nice study, and I congratulate the author on this. Overall, the data supports the paper's claim that metazoan evolution of the oxygen-sensing pathway involved conserved divergence of VHL affinity for HIF1 α and HIF2 α .

While I am supportive of publication in principle in the journal, a weakness of the current version is that the biophysical data presented is not of adequate quality to support the important conclusions on the relative differences in binding affinity and in particular dissociation kinetics. These differences are clearly subtle, for example 2-3 fold overall between HIF-1a and HIF-2a due to the Met-Leu change. So it is absolutely paramount that the best, most sensitive and highest quality of biophysical measurements is obtained. Unfortunately, the BLI data obtained is problematic. Visual inspection of the BLI data presented (Figures 2C and Figure S2) shows clearly that the curvatures of the fitting curves (both in the association and dissociation phases) depart significantly from the curvature of the observed data. This suggests that the simple 1:1 binding model is not adequate to fit the data, most likely due to mass transport limitation, re-binding or other artefactual phenomena to do with the technology. A simple plot of the residuals (difference between data and fit) as a function of time for each concentration curves will visually and clearly illustrate the problem. A model that adequately fits the data should yield residuals that are distributed randomly. Instead, when residuals are non-random, and instead follow a pattern, that strongly suggests that the model is not adequate to fit the data. Inspection of the data included in the paper suggests that the residuals (once shown) would prove to be non-random. This is perhaps not surprising given that the data was recorded with the Blitz instrument from ForteBio, which is really their entry instrument and best suited for quick checks of protein presence in crude matrices as well as at various stages of purification, and perhaps high affinity protein-protein interactions such as antibodies. Also it is widely accepted in the field that SPR is much superior than BLI to resolve kinetics. The authors are strongly encouraged to utilise an alternative biophysical technique to measure binding affinities and dissociation off-rates. I would strongly recommend SPR e.g. Biacore 3000 or even better T100/200, which would guarantee much more robust data and conclusions.

Response: We thank the reviewer for the careful evaluation of our BLI data. Indeed, although the residual values are uniformly small, they do follow a non-random distribution for the various concentrations of VHL complex (see **New Fig. X for Reviewer**). Thus, we followed the suggestion to validate our BLI kinetic data using SPR. We collected kinetic data on the key interaction of Bio-HIF1 α (WT), Bio-HIF1 α OH(WT), Bio-HIF2 α OH(WT), and Bio HIF1 α OH(M561T) with VHL complex(WT) using Biacore X100. Consistent with our BLI results, we show that HIF α Met_{n-3} does increase complex stability with VHL, as indicated by a lower dissociation rate (**New Supplementary Fig. 2b, c**). Moreover, as the SPR experiments were conducted at 25 °C (BLI experiments involved incubating proteins and buffer on ice before use), the dissociation rates were generally faster but the trend was similar as observed using BLI.

Fig. X. Residuals of raw and fitted biolayer interferometry data. The residual between the observed, raw data and the fitted data (1:1 global model) is graphed as a function of time and VHL complex concentration. Values represent mean \pm SEM.

Other minor points:

1. Figure 2C: confusing to show representative BLI data with HIF1 α -OH and HIF-2 α (presumably Pro instead of Hyp) – as it could be misleadingly interpreted as HIF1 α peptide binds while HIF2 α peptide does not. To avoid this, the authors should show either data for the same HIF e.g. HIF-1, in one case Hyp in the other case Pro; alternatively, show data for both HIF-1 α -OH and HIF-2 α -OH.

Response: It is well established that unhydroxylated HIF α , whether HIF1 α or HIF2 α , does not bind VHL. We simply included unhydroxylated HIF2 α as a negative control. To avoid any potential confusion, we moved the sensorgrams to **Supplementary Fig. 2a** with the other sensorgrams.

2. “Although oxidation of Met561 and Met568 resulted in a significant increase in the dissociation rate of VHL from HIF1 α OH peptide, only oxidation of Met561 resulted in a decrease in dissociation rate comparable to the M561T substitution (Fig. 4d, e).” please revise to ensure

increase vs decrease of dissociation rate are described correctly.

Response: Done.

3. The authors might want to streamline the Discussion to make it more concise and clear.

Response: Although we appreciate this comment, we also had to respect and satisfy the requests made by Reviewer 2, which required expansion of the discussion. We have, however, made every effort to keep any expanded discussion as concise and clear as possible.

REVIEWER No. 2

This manuscript of Tarade et al report how the divergence hypoxia inducible factors HIF1alpha and HIF2alpha involved specific amino acid substitutions that affect their affinity to von Hippel-Lindau (VHL) tumor suppressor protein. This in turn partly may contribute to the observed difference of HIF1alpha and HIF2alpha in moderate oxygen tensions, HIF2alpha having weaker interaction with VHL and higher stability. Previously, HIF fragments containing the parts of the oxygen dependent domain including immediate surrounding of the critical proline have often been studied in the context of their interaction with prolyl hydroxylases that require oxygen as a substrate for the reaction. In this study the HIF fragments are used to study the HIF interaction with VHL that tags HIF to degradation after hydroxylation of the critical proline. The key novelties of the paper include the combination of phylogenetically informed selection of diverged amino acid positions for series of mutations for kinetic measurements, and the finding that the mutation of HIF1 α Metn-3 to “HIF2 α type” Thr has a significant effect on the affinity of VHL for the HIF peptide. This finding is of general interest for the field of hypoxia research, but also evolutionary biology. When considering the evolution of the three components of the HIF system: HIFs and PHDs have both 3 paralogs in mammals whereas VHL only one – this study highlights that the interactions with the non-duplicated component of the system has been important in the evolution of the complex as a whole. The methodology, presentation and reporting are convincing, and in key experiments three independent repeats were conducted. The figures and supplementary data are clear and the methods are described sufficiently. In the discussion part the argumentation is partly coupled to a recent conceptual model introduced by Hammarlund et al 2018, but the results of this paper have limited applicability to substantiate this model. However, overall, this paper advances our understanding of the evolution of HIF pathway and the divergence of HIF alpha paralogs and is a valuable scientific contribution.

1. In the discussion the authors focus on how HIF CODD Metn-3 influences VHL interaction when looking at the evolution of HIF pathway. The authors should add a paragraph to discuss the new findings also in the light of relative contribution of HIF-VHL interactions versus HIF-PHD interactions in the evolution of HIF mediated oxygen-sensing. Also, the combined contribution of NODD and CODD to the VHL interaction could be touched on. Further, invertebrates encode a variable sets of HIF components in their genomes that based on their conservation are predicted to have functional importance. For example, roughly, placozoa has only CODD and one PHD, cnidarians have only CODD and two PHDs, and chordates both NODD and CODD as well as two PHDs. How does the variation in the discussed Metn-3 and VHL correlate with the presence of multiple ODD or PHDs?

Response: We thank the reviewer for these comments. We have included at the end of the first paragraph a discussion of other relevant HIF-PHD-VHL evolutionary trends, such as the duplication of an ancestral ODD to yield two ODD sites and the duplication of an ancestral PHD gene. Both of these evolutionary events (ODD and PHD duplication) became fixed in the metazoan lineage before the HIF α Metn-3/VHL Phen+3 combination became fixed. See **lines 335-348**.

2. Authors should comment on how the experiments using short HIF peptides (now 19 amino acids) present a reasonable approximation of the whole HIF CODD region or the HIF protein interaction with VHL.

Response: Done. See lines 390-400.

3. Design for Figures 3c, 5a, 6a and 6b. Showing variable positions of a-a s with % bars is useful for the vertebrate vs invertebrate comparison, but not useful for looking at ancestral states among invertebrates. The figure with phylogeny is brought in maybe a bit too late in 6b, I suggest that 6b could be introduced earlier and in a more reader friendly format. The current % figs 5a 6a could be in the supplementary, and for the main figure(s) it might be better to modify 6b tree that focuses on invertebrates, so that you depict the pattern of crucial amino acids in HIF and VHL (and possible also the branch specific dn/ds) on the branches of the tree - not just alignment. Using some ancestral state construction method (see Table 1 in Joy JB 2016 PLoS Comput Bio) would substantiate the subject that you present on page 6. " By constructing an idealized phylogeny on the basis of well-characterized branching points during metazoan evolution (radial symmetry versus bilateral symmetry, protostome versus deuterostome), several possible evolutionary trends are possible. The simplest model suggests that the ancestor to extant deuterostome species possessed a combination of VHL Phe_{n+3} and HIF1 α Met_{n-3}...."

Response: We appreciate these comments, which we believe improve the clarity of the manuscript. We agree that including the specific HIF α X_{n-3} substitutions (**Fig. 3c**) is unnecessary and makes the figure busier. They have been excluded. We have also moved (former) **Fig. 5a** to the supplementary data (now **Supplementary Fig. 4a**). However, we believe it provides clarity to include the specific VHL substitutions that are prevalent in invertebrate species. We have also moved **Fig. 6a** to the supplemental data (now **Supplementary Fig. 4b**). Regarding panel **6b**, we have included it in **Fig. 5f**. We found that it was difficult to include this information earlier in the manuscript as the idealized phylogeny serves as a summary of our studies on both HIF1 α and VHL substitutions. Further, focusing solely on invertebrates would ignore the divergence of HIF2 α in the vertebrate lineage. Thus, we have largely kept this panel the same, and included the VHL sequence data for *T. Adhaerens* (**Revised Fig. 5f**)

We performed an ancestral reconstruction analysis (Maximum parsimony) analysis utilizing our HIF1 α alignment and phylogenetic tree (**New Supplementary Fig. 7**). The analysis suggested that the last common ancestor of deuterostomes and protostomes possessed HIF α Met_{n-3}, which is earlier than we predicted on the basis of manual inspection. Unfortunately, despite several attempts, we were unable to produce a suitable phylogenetic tree (based on VHL sequences) that would allow for ancestral VHL sequence reconstruction. Overall, we have tempered our conclusion and instead discussed multiple possibilities for when the combination HIF α Met_{n-3} and VHL Phe_{n+3} may have emerged. See lines 254-262.

4. Page 7. " Conversely, among deuterostome species we instead observe that VHL Phe_{n+3} and HIF1 α Met_{n-3} are conserved (Fig. 6b). Our biochemical and biophysical experiments would suggest that HIF1 α /VHL complex stability is increased in deuterostome invertebrates, from which vertebrate species evolved, compared to radiata and protostome species (Fig. 5d, e, f). Upon the evolution of vertebrate species, the ancestral HIF1 α gene..."
- In invertebrates there is HIF α not HIF1 α .

Response: Corrected.

5. Page 8. "One emergent model of hypoxia sensing and metazoan evolution focuses on the propensity for oxic niches to promote cell differentiation (29). By extension, rather than HIF α evolving as a mechanism to allow cells to survive low oxygen states, HIF α might have been necessary for the maintenance of stem cell populations under high oxygen conditions."

- Please, directly state that now the results will be discussed with emphasis on this (new) model. Also, if possible, directly state if the results presented in this paper support this model. The model has quite reasonable arguments, but it remains a bit unclear if it is associated with a testable hypothesis/predictions. Maybe it would be useful to add short description on if and how your results fit better the "pseudohypoxia model" rather than its simplistic "counterargument". "Counterargument" or old line of thought being (here just roughly simplified), that in invertebrate deuterostomes the increased affinity of VHL-HIF would present pressure towards tighter hypoxia dependent regulation of HIF and then in vertebrates the selection on HIF1 and HIF2 lead to wider range of stabilization in different oxygen tension to enhance more fine-tuned hypoxia responses.

Response: We have a statement stating that we will be discussing our results in the context of the 'pseudohypoxic' model. See **lines 369-370**. From our understanding of the pseudohypoxic model, our hypothesis that deuterostome species more tightly regulate HIF α signalling (corollary that radiata and protostome species weakly regulate HIF α signalling) is consistent with a 'pseudohypoxic' model. However, we added a testable prediction that can clarify these two arguments. If pseudohypoxic maintenance is an important HIF α role in invertebrate species, we would expect that HIF α protein should be present under ambient atmospheric conditions, much like HIF2 α in mammalian cells. See **lines 379-382**.

Minor revisions:

Page 3. In addition to (17), add Smythies JA et al (2019) for chromatin binding.

Response: Done.

Page 3. "Despite well-documented differences between HIF1 α and HIF2 α , little is known about how processes of natural selection acted upon the coding region of these two genes to optimize the cellular response to changes in oxygen during evolution of the metazoans."

- do not make a paragraph out of one sentence.

- In invertebrates there is HIF α and in vertebrates HIF1 α and HIF2 α , thus instead of metazoans in this sentence structure it should be vertebrates.

- Modify, for example mention that natural selection on these two genes has only been studied using computational sequence analysis (such as Rytönen KT et al 2011) but the role of specific amino acid substitutions have not been assessed by kinetic measurements of protein function...

Response: Done. See **lines 101-102**.

Page 3. "We have previously noted that in the immediate vicinity of the primary hydroxylation sites..." - Add a reference

Response: Done.

Page 4. ... Fig 3b, modify the figure so that the readers can immediately see when NODD changes to CODD, for example add a space,
Below the figure, for clarity, mark hydroxyprolines with red.

Response: Done.

Page 8. "However, as noted above, studies with full-length VHL and HIF α proteins from representative invertebrate species would be required to further speculate on the evolution of oxygen-sensitivity in HIF α proteins."

- also consider that PHDs are important for the system, and invertebrates have variable numbers PHD gene copies and HIF ODDs.

Response: Done. See **lines 335-348**.

Page 8. "One advantage of vertebrate hypoxia-sensing is the expanded repertoire of hypoxia sensitive transcription factors allowing specialization"

- This is an odd sentence, modify, for example: early vertebrate genome duplications also contributed to the diversification and specialization of HIF pathway

Response: Done.

Page 9. "Furthermore, these substitutions appear to affect only one paralog, resulting in an increasingly fine-tuned hypoxia response in vertebrate species, where one species is stabilized over a wider range of oxygen tensions and may be particularly important for pseudohypoxia maintenance."

- species ? = gene paralog

Response: Done.

REVIEWERS' COMMENTS:

Reviewer #1 (Remarks to the Author):

The authors have made a good attempt to incorporate additional data and address the suggested changes based on the initial round of review; this has improved the paper. In particular the new SPR data provide valuable orthogonal validation of found trends in binding affinity and kinetics. I am supportive of publication in the current form.

Reviewer #2 (Remarks to the Author):

Review 2, round 2:

Overall the authors have conducted the revisions that I requested and I recommend the paper to be accepted for publication. Authors can be trusted to implement following minor revisions without additional reviewer round:

Minor revisions:

Lines 253-276: Following the arguments on different invertebrate groups is difficult when reading the text and looking at Fig S7 and Fig 5F. Add higher taxonomic order names like "radiata and bilateria" and "deuterostome and protostome" etc. to Fig S7. Then for Fig 5F consider adding also exemplary species names either to the left of right of the aa sequences (Fig. 5E box-blots can be thinner).

Lines 280-282 "we constructed an idealized phylogeny on the basis of well-characterized branching points during metazoan evolution (radial symmetry versus bilateral symmetry, protostome versus deuterostome)."

- Add (Fig. 5f.)... It is confusing that you reference 5f on line 272 and then introduce it here "as new", consider moving this sentence up.

Lines 311-456 Discussion

Thank you for including the additional points I suggested. Anyhow, I would also echo Reviewer 1's comment on streamlining the discussion more concise and clear. Please consider, for example, cutting the first paragraph (312-348) into two separate ones and making the second one (349-382) shorter.

REVIEWERS' COMMENTS:

Reviewer #1 (Remarks to the Author):

The authors have made a good attempt to incorporate additional data and address the suggested changes based on the initial round of review; this has improved the paper. In particular the new SPR data provide valuable orthogonal validation of found trends in binding affinity and kinetics. I am supportive of publication in the current form.

Response: We reiterate our appreciation for the helpful advice provided.

Reviewer #2 (Remarks to the Author):

Review 2, round 2:

Overall the authors have conducted the revisions that I requested and I recommend the paper to be accepted for publication. Authors can be trusted to implement following minor revisions without additional reviewer round:

Minor revisions:

Lines 253-276: Following the arguments on different invertebrate groups is difficult when reading the text and looking at Fig S7 and Fig 5F. Add higher taxonomic order names like “radiata and bilateria” and “deuterostome and protostome” etc. to Fig S7. Then for Fig 5F consider adding also exemplary species names either to the left of right of the aa sequences (Fig. 5E box-blots can be thinner).

Response: We have added exemplary species to the right of the aa sequences. Further, we labeled higher taxonomic orders for Supplementary Figure 7.

Lines 280-282 “we constructed an idealized phylogeny on the basis of well-characterized branching points during metazoan evolution (radial symmetry versus bilateral symmetry, protostome versus deuterostome).”

- Add (Fig. 5f.)... It is confusing that you reference 5f on line 272 and then introduce it here “as new”, consider moving this sentence up.

Response: We have labelled lines 280-282 (now lines 286-289) with Fig. 5f. We no longer reference Fig. 5f on line 272 (now line 279) and instead reference Supplementary Figure 3, which contains actual sequence data.

Lines 311-456 Discussion

Thank you for including the additional points I suggested. Anyhow, I would also echo Reviewer 1’s comment on streamlining the discussion more concise and clear. Please consider, for example, cutting the first paragraph (312-348) into two separate ones and making the second one (349-382) shorter.

Response: We have reworked the first paragraph, having divided it into three smaller paragraphs that we believe is clearer.